# FOXO1 Confers Maintenance of the Dark Zone Proliferation and Survival Program and Can Be Pharmacologically Targeted in Burkitt Lymphoma

**DOI:** 10.3390/cancers11101427

**Published:** 2019-09-25

**Authors:** Franziska Gehringer, Stephanie E Weissinger, Lotteke JYM Swier, Peter Möller, Thomas Wirth, Alexey Ushmorov

**Affiliations:** 1Institute of Physiological Chemistry, University of Ulm, 89081 Ulm, Germany; franziska.gehringer@uni-ulm.de; 2Institute of Pathology, University of Ulm, 89081 Ulm, Germany; weissinger.stephanie@googlemail.com (S.E.W.); peter.moeller@uniklinik-ulm.de (P.M.); 3Department of Pathology and Medical Biology, University of Groningen, 9747 AG Groningen, The Netherlands; l.j.y.m.swier@umcg.nl

**Keywords:** FOXO1, Burkitt lymphoma, germinal center dark zone B cell proliferation program

## Abstract

The FOXO1 transcription factor plays a central role in the proliferation and survival of B cells at several stages of differentiation. B cell malignancies, with exception of classical Hodgkin lymphoma, maintain expression of FOXO1 at levels characteristic for their non-malignant counterparts. Extensive expression profiling had revealed that Burkitt lymphoma (BL) show many characteristics of the dark zone (DZ) germinal center (GC) B cell program. Here we show that FOXO1 knockdown inhibits proliferation of human BL cell lines. The anti-proliferative effect of the FOXO1 knockdown is associated with the repression of the DZ B cell program including expression of MYB, CCND3, RAG2, BACH2, and CXCR4. In addition, the induction of signaling pathways of the light zone (LZ) program like NF-κB and PI3K-AKT was observed. Using a rescue experiment we identified downregulation of the proto-oncogene MYB as a critical factor contributing to the antiproliferative effect of FOXO1 knockdown. In an attempt to estimate the feasibility of pharmacological FOXO1 repression, we found that the small molecular weight FOXO1 inhibitor AS1842856 induces cell death and growth arrest in BL cell lines at low concentrations. Interestingly, we found that overactivation of FOXO1 also induces growth inhibition in BL cell lines, indicating the importance of a tight regulation of FOXO1 activity in BL.

## 1. Introduction

Burkitt lymphoma (BL) is an aggressive B cell lymphoma [1] which originates from the germinal center (GC) [2], and is characterized by oncogenic translocations of the proto-oncogene *MYC* [3]. The GC consists of two main histological and functional compartments known as dark zone (DZ) and light zone (LZ). In the DZ, B cells undergo somatic hypermutation and actively proliferate and afterwards move to the LZ where they receive survival signals via the B cell receptor (BCR) and CD40 in case of successful recombination and expression of a high affinity antibody. The DZ gene expression program depends on expression of CCND3 and the transcription factors BCL6, FOXO1, and TCF3. In contrast, the DZ program is repressed by BCR and CD40 signaling [4] in LZ B cells. At the same time, signaling from the BCR and CD40 [4], which activate NF-κB, JAK-STAT, ERK, and PI3K-AKT pathways, is essential for survival and further differentiation of the LZ B cells [5,6,7].

Although MYC translocation under the control of immunoglobulin loci is an essential oncogenic event, it is not sufficient for BL progression. The maintenance of the main components of the DZ program [8] including physiologically high expression of FOXO1 [9] and TCF3 [1,10] is essential for BL. Moreover, activating mutations of TCF3 and FOXO1, inactivating mutations of TCF3 antagonist ID3, and protein stabilizing mutations of CCND3 belong to the most frequent oncogenic events in BL [10,11,12].

The FOXO family of transcription factors regulates multiple processes, including cell cycle progression, apoptosis, glucose metabolism, differentiation, protection from oxidative stress, and stem cell maintenance [13,14,15]. In some B cell malignancies, FOXO1 acts as a tumor suppressor and its activation induces growth arrest and apoptosis [13,16,17,18]. Surprisingly, FOXO1 knockdown in the MYC-PI3K driven mouse model of BL resulted in cell death and growth arrest [9]. Moreover, *FOXO1* gene editing results in time-dependent selection of in-frame edited clones [9] and impedes proliferation of BL cell lines [19], indicating a role of FOXO1 in BL lymphomagenesis.

Using gene expression profiling (GEP), we found that FOXO1 knockdown, *inter alia*, represses DZ signature genes and this was associated with the activation of LZ signaling such as activation of PI3K-AKT and IKK-NF-κB pathways. Moreover, we demonstrated the feasibility of pharmacological inhibition of FOXO1 to mimic genetic FOXO1 down-regulation in BL as a potential therapeutic strategy.

## 2. Results

### 2.1. FOXO1 Expression is Essential for BL Survival

BL tumors maintain high levels of FOXO1, comparable to normal CD19^+^ B cells [20]. Since the activity of FOXO transcription factors is regulated by changing the nuclear localization of FOXOs, we used subcellular fractionation (Appendix A) and immunofluorescent staining of FOXO1 in BL samples and cell lines to investigate whether FOXO1 is potentially active (Appendix A, respectively). Given that occasional monoallelic mutations in an N-terminal hotspot were suggested to confer FOXO1 insensitivity to AKT-dependent phosphorylation in BL, we included cell lines harboring both wild type (Ramos, BL-41, Daudi, Raji) and also mutated FOXO1 alleles (Namalwa, Jiyoye) (Appendix A). Both methods identified a substantial fraction of FOXO1 in the nuclei, independently of its mutational status, supporting a role of active FOXO1 in the oncogenic program of BL.

Genomic editing of BL cell lines was shown to increase positive selection of in-frame edited clones in comparison to out-of-frame edited clones [9]. To find out whether acute depletion of FOXO1 has an antitumor effect in BL cell lines, we transduced BL cells with vectors expressing the fluorescent marker RFP and shRNAs targeting different *FOXO1* sites (Figure 1A) and monitored the dynamic of the RFP^+^ population (Figure 1B). F1sh specifically targeted *FOXO1*, whereas Fsh1/3/6 might in addition target *FOXO3* and *FOXO6*, but preferentially downregulated *FOXO1* expression [21,22]. The cHL cell lines L428 and U-HO1, which do not depend on FOXO1 [20,23], were used as negative controls. In all BLs, both shRNAs decreased the proportion of RFP^+^ cells in comparison to cells transduced with the scrambled shRNA, independently of the *FOXO1* mutational status (Appendix A). In contrast, the cHL cell lines were insensitive to *FOXO1* knockdown. In addition, we corroborated our results on the antitumor effect of *FOXO1* knockdown using CRISPR/Cas9 genome editing (Figure 1C and Appendix A).

To understand the mechanisms of the low performance of the FOXO1-depleted cells in the competitive proliferation assay, we analyzed the cell cycle distribution and induction of apoptosis. In two cell lines, *FOXO1* knockdown induced G_1_-arrest, whereas in Ramos it slightly increased apoptosis (Figure 1D,E). Finally, to prove the specificity of the anti-proliferative effect of *FOXO1* knockdown, we co-expressed the *FOXO1* shRNA with a wild type *FOXO1* harboring a wobbled F1sh target site (F1wob) in BL-41 and Ramos. We found that the expression of F1wob resulted in a significant protection of the cells from the knockdown of endogenous FOXO1 (Figure 1F,G). 

### 2.2. FOXO1 Knockdown Inhibits the Expression of Core-Survival Genes in BL 

To understand the molecular mechanisms responsible for the antiproliferative effect of FOXO1 depletion in BL, we used gene expression profiling (GEP). Using a Human Gene 1.0 ST Affymetrix array, we first identified the top 20 genes concomitantly downregulated in BL-41 and Namalwa by the FOXO1 knockdown. Among them were the known FOXO1 targets *CXCR4* [5] and *RAG2* [24] as well as critical regulators of BL proliferation *MYB*, *MAD2L1*, and *BUB1* [25,26,27,28] (Figure 2A and Appendix A). Using qRT-PCR we validated the downregulation of *MYB* as well as FOXO1 target genes *CCND3* [29,30], *IRF4* [31], *BACH2* [18,30,32], and DNA repair genes *RAD51* and *RAD51AP1* (Figure 2B). 

Given that CCND3 contributes to the oncogenic program of BL [1], we analyzed its expression at the protein level after FOXO1 knockdown, but failed to reveal any modulation (Appendix A). Moreover, co-transfection of CCND3 did not rescue BL-41 and Namalwa cell lines from the anti-proliferative effects of *FOXO1* knockdown (Appendix A), indicating that repression of CCND3 is not the key event for the anti-proliferative effects observed after FOXO1 knockdown. Given a role of FOXO1 in the protection against reactive oxygen species [14], we analyzed oxidative stress levels and expression of the mitophagy marker TOM20 after *FOXO1* knockdown, but did not observe any effects either (Appendix A).

Using GSEA (Appendix A) we identified proliferation-associated signatures, including cell cycle related targets of E2F transcription factors, G2/M checkpoint, and mitotic spindle assembly signatures (Appendix A). Furthermore, we detected a repression of FOXO1 target genes in GC B cells signature [5] (Appendix A) upon *FOXO1* knockdown. The signatures of core-survival BL genes [33] (Appendix A) including *MYB* [27] (Appendix A) (Figure 2C) were also significantly repressed. We validated MYB downregulation at the protein level (Figure 2D) and corroborated the previously shown essential role of MYB in BL maintenance [26,27] using shRNA knockdown (Figure 2E,F). Finally, we confirmed the role of MYB repression in the antitumor effect of FOXO1 depletion by a rescue experiment. Overexpression of the full-length 100 kDa version of *MYB*, which is expressed by BL cells in addition to the 75 kDa splice variant and shares similar properties [34], protected BLs from the antiproliferative effect of *FOXO1* knockdown (Figure 2G,H) by relieving the G_1_-arrest (Appendix A). 

Thus, FOXO1 is essential for the expression of core survival genes in BL including *MYB*.

### 2.3. FOXO1 Is Critical for Maintenance of the DZ Phenotype and for Control of the NF-ΚB and PI3K-AKT Activity in BL 

Given an essential role of FOXO1 in maintenance of the DZ program [1,6], we applied gene expression signatures of human tonsillar DZ (CXCR4^+^) and LZ (CXCR4^−^) cells [36,37] to the FOXO1 knockdown signature. We found that FOXO1 depletion significantly repressed the DZ signature and enriched the LZ signature (Figure 3A and Appendix A; Appendix A). Since BL depends on the DZ survival and proliferation program [1,2,37,38], we analyzed the expression of the DZ marker CXCR4 after FOXO1 repression by flow cytometry (Figure 3B). We observed a strong downregulation of CXCR4 in cells transduced with the FOXO1 shRNA, but not in cells expressing the control vector. In addition, the isotype control was not altered. Given that the decrease of CXCR4 on the cell surface measured by the fluorochrome-coupled antibody might be explained by activation-induced CXCR4 internalization [39], we analyzed CXCR4 expression by immunoblot. We found a strong decrease of CXCR4 expression. This supports the concept of transcriptional repression as the important mechanism (Figure 3C). Finally, to exclude the possibility that the observed downregulation of GC DZ genes represents an shRNA off-target effect we corroborated the results of FOXO1 knockdown by FOXO1 genome editing. Using immunoblot, we found that both sgRNAs downregulated the expression of CXCR4 and MYB in the BL cell line Namalwa (Appendix A).

As activation of NF-κB contributes to the LZ program [37], we applied a “NF-κB activation in BL” signature [40] and found its significant enrichment after *FOXO1* knockdown (Figure 3A and Appendix A). NF-κB activation was proven by an increase of IKK-dependent RELA/p65 phosphorylation at the S536 residue (Figure 3D) and by induction of a NF-κB-responsive luciferase reporter construct (Figure 3E). 

PI3K-AKT activity is also higher in LZ than in DZ B cells [5,6]. We therefore analyzed the AKT phosphorylation status and observed an increase of the pAKT^T308^ signal by the FOXO1 knockdown (Figure 3F). We conclude that FOXO1 contributes to the maintenance of the DZ program by inducing CXCR4 and MYB and via attenuation of the IKK-NF-κB and AKT activity in BL. 

### 2.4. Pharmacologic FOXO1 Inactivation Inhibits Proliferation and Induces Apoptosis in BL Cell Lines

Recently, we revealed the cytotoxic activity of a small molecular weight FOXO1 inhibitor AS1842856 against BCP-ALL in *in vitro*, *ex vivo*, and *in vivo* models [21]. Therefore, we measured the sensitivity of BL cell lines to the inhibitor (Figure 4A). As sensitivity burden we chose 300 nM, which is the plasma concentration of AS1842856 in mice after oral administration in an experimental anti-diabetic treatment [42]. This concentration has been proven to have a robust antitumor effect against BCP-ALL patient-derived xenotransplants *in vivo* [21]. Four of six BL cell lines were sensitive to the inhibitor (IC_50_ values between 2 and 94 nM). Moreover, AS1842856 induced CASP3 cleavage (Figure 4B), which was associated with apoptosis in two of three cell lines (Figure 4C) and G_1_-cell cycle arrest (Figure 4D). The inhibitor downregulated transcription of FOXO1 targets similar to *FOXO1* knockdown (Figure 4E). AS1842856 also induced RELA^S536^ phosphorylation and NF-κB-dependent transcription (Figure 5A,B). Moreover, it downregulated MYB expression and induced AKT^T308^ as well as AKT^S473^ phosphorylation (Figure 5A,C, respectively). Similar to *FOXO1* knockdown using the shRNA, AS1842856 treatment resulted in a strong CXCR4 repression (Figure 5D,E). In contrast to the genetic FOXO1 repression, pharmacologic FOXO1 inactivation also repressed CCND3 at the protein level, independently of its CCND3 mutational status (Figure 5F and Appendix A). Interestingly, MYB overexpression completely rescued BL cells from the cytotoxic effect of the inhibitor (Figure 5G,H), indicating that MYB downregulation is a crucial factor in the toxic effect of AS1842856.

Surprisingly, and in contrast to the genetic knockdown, AS1842856 did not inhibit *MYB* transcription (Figure 6A). The downregulation of MYB at protein level could further not be explained by the increase of protein degradation (Figure 6B). It had been shown that MYB levels are attenuated by miR-150, a micro RNA downregulated in BL [43]. Thus, we explored the influence of AS1842856 on the expression of miR-150 and found that AS1842856 strongly induced miR-150 in BL cell lines (Figure 6C). With help of a CRISPR/Cas9-mediated *miR-150* knockout construct, we demonstrated that *miR-150* knockout protects BL cell lines from AS1842856-induced MYB downregulation (Figure 6D–F), indicating that AS1842856 downregulates MYB proteins levels by inducing the MYB-targeting miR-150. At present, the divergent mechanism by which genetic and pharmacological interference with FOXO1 affect MYB are not completely understood.

Nevertheless, pharmacologic FOXO1 inhibition mimics the effects of genetic *FOXO1* downregulation.

### 2.5. Genetic FOXO1 Hyperactivation is Toxic for BL

Previous studies by us and others had indicated that FOXO1 activity needs to be maintained in a strictly regulated range and both, too little as well as too much is bad for these tumors (Goldilocks principle) [21,23]. Given that genetic and pharmacological activation of *FOXO1* induce apoptosis in different B cell lymphomas [13,16,17,18], we overexpressed an AKT-resistant inducible version of *FOXO1* (FOXO1(3A)ER) (Figure 7A) in BL-41 and Namalwa BL cell lines. FOXO1(3A) ER activation inhibited the growth of BL cell lines (Figure 7B), accompanied by induction of apoptosis and G_1-_cell cycle arrest (Figure 7C,D, respectively). Consistently, we observed an upregulation of the pro-apoptotic FOXO1 target gene *TNSF10/TRAIL* (Figure 7E), CASP3 cleavage, and increase of cyclin-dependent kinase inhibitor CDKN1B (Figure 7A). Importantly, overexpression of wild type FOXO1 also inhibited the growth of BL cell lines (Figure 7F,G).

We conclude that both down- and up-regulation of FOXO1 activity negatively influence the survival and proliferation of BL, indicating that, similar to BCP-ALL [21], FOXO1 acts as a rheostat in the oncogenic program of this utmost aggressive lymphoma. 

## 3. Discussion

In the present work, we show that acute genetic depletion of FOXO1 inhibits the proliferation of BL cell lines. In particular, we observed that repression of FOXO1 activated signaling pathways characteristic for the GC LZ program, like the PI3K-AKT and IKK-NF-κB pathways. This ultimately downregulated the GC DZ program and DZ markers like CXCR4. We identified down-regulation of the proto-oncogene MYB as an important factor contributing to the anti-proliferative effect of FOXO1 knockdown. Importantly, pharmacological repression of FOXO1 also induced cell cycle arrest and apoptosis in BL cell lines and partially reproduced the effects of the shRNA-mediated FOXO1 knockdown on gene transcription. Finally, we demonstrated that overactivation of the FOXO1 activity also induces cell death and growth arrest, indicating the importance of a tight regulation of FOXO1 activity for the survival of BL.

We have shown that knockdown of FOXO1 in BL cell lines either interferes with cell proliferation or the apoptotic program. Recently it has been shown that knockdown of FOXO1 in a MYC/PI3K hyperactivation-driven model of BL in mice induces apoptosis. In BL cell lines, genetic editing of the FOXO1 gene resulted in positive selection of in-frame edited clones, indicating a role of FOXO1 in BL [9]. Therefore, our data support the negative effect of FOXO1 depletion on BL proliferation, but at the same time indicate differences between BL cell lines and the mouse model, especially at the molecular level.

Interestingly, gene expression profiling of BL had revealed the virtual absence of NF-κB activity [38] and the characteristic expression of markers of the GC DZ program [2,37,44]. We show up-regulation of PI3K-AKT and NF-κB activity in BL cell lines by FOXO1 depletion. This is in line with higher PI3K-AKT activity in LZ B cells, which also express lower FOXO1 levels than DZ B cells [5,6]. FOXO1 has a complex effect on PI3K-AKT activity. Even though PIK3CA, the catalytic subunit of PI3K, was identified as a transcriptional target of FOXOs [45], FOXO1 suppresses PI3K activity in NSCLC cell lines, although the mechanism is unknown [46]. 

It is conceivable that activation of IKK-NF-κB contributes to PI3K-AKT activation. The NF-κB signature is repressed in BL in comparison with ABC- and even with GC-DLBCL [44]. Moreover, NF-κB activation represses MYC-driven lymphomagenesis and is toxic for BL cell lines [40,47]. Since FOXO3A was shown to inhibit NF-κB activation [48] by direct binding to RELA [49], it is conceivable that the structurally and functionally related FOXO1 protein acts in a similar way. Being activated, AKT might also contribute to NF-κB activation by mTORC1-dependent IKK phosphorylation [50]. NF-κB, in turn, can potentiate PI3K-AKT activity, e.g., by suppression of PTEN [51], creating a self-amplifying circuit. Of note, the NF-κB activating kinases IKKs can inactivate FOXO proteins [52,53], suggesting that the efficacy of the FOXO1 depletion might be further increased by IKKs. Thus, FOXO1 acts as a regulator of NF-κB and PI3K-AKT activity in BL and its inhibition results in auto-amplification of IKK-NF-κB and PI3K-AKT pathways leading to repression of the DZ program. 

In accordance with up-regulation of IKK-NF-κB and PI3K-AKT activity we observed repression of the DZ and up-regulation of the LZ signatures in BL by FOXO1 knockdown. In fact, we saw similar negative effects of *FOXO1* knockdown in BL cell lines as previously described for DZ B lymphocytes [6]. Although downregulation of the DZ program by *FOXO1* knockdown in a MYC-driven mouse B cell lymphoma model that harbored a constitutively active version of the PI3K catalytic subunit has been recently reported [9], the repertoire of repressed genes was different from what we obtained in human cell lines. Among others, *MYB*, *CCND3*, and *CXCR4* were not downregulated in the mouse model [9]. We consider repression of the DZ program as the main cause for the cell cycle arrest induced by *FOXO1* knockdown.

Due to the complexity of the DZ program, it is difficult to filter out a single factor responsible for the growth arrest. Nevertheless, we identified repression of MYB as an important anti-proliferative event. Of note, MYB is a core BL survival factor, which is highly expressed in centroblasts [26,27]. Importantly, although *MYB* is repressed by *FOXO1* knockout in mouse GC B cells [6], the binding of FOXO1 to the *MYB* promoter was not detected by ChIP-sequencing neither in GC B cells [5], nor in other tissues [30] indicating involvement of other mechanisms (e.g., protein-protein interactions) [54]. Indeed, when we analyzed the effect of pharmacological FOXO1 inhibition we found that induction of miR-150 plays a critical role in regulating MYB levels. In contrast, CXCR4 is a direct FOXO1 transcriptional target [5], which is downregulated by FOXO1 depletion in normal GC B cells [36,37]. Remarkably, we also find a strong repression of CXCR4 by FOXO1 downregulation. Since CXCR4 is also a MYB target [55,56,57], MYB downregulation might potentiate FOXO1-inactivation induced CXCR4 repression.

We found that the small molecular weight FOXO1 inhibitor AS1842856 is toxic for BL. Although the specificity of small molecular weight inhibitors is a matter of concern in general, the treatment with AS1842856 reproduced most effects of the genetic FOXO1 knockdown, including repression of DZ-specific genes and growth inhibition. Interestingly, most BL cell lines were as sensitive to the inhibitor as BCP-ALL cell lines [21], indicating the existence of common oncogenic mechanisms acting in these B cell neoplasia. 

The most obvious differences between shRNA-mediated and pharmacological FOXO1 inactivation were the inability of AS1842856 to repress MYB at mRNA level. Interestingly, repression of MYB protein expression was due to induction of *miR-150* transcription. These discrepancies might be explained all by differences in the mechanisms of action. RNA interference decreases the FOXO1 levels, whereas AS1842856 does not modulate FOXO1 expression or localization, instead, AS1842856 binds to the transactivation domain and thereby *bona fide* interferes with the FOXO1 transactivation activity [42]. It is conceivable that not all interactions of FOXO1 with other proteins might be blocked by AS1842856, moreover the binding specificity of the new structures of AS1842856 might be different. Importantly, AS1842856 binds, although to a lesser extent, to other members of the FOXO family, the effects of whose are only partially redundant [42]. Inhibition of other FOXOs might explain the generally stronger effects of pharmacological in comparison to genetic FOXO1 inhibition in regard to gene expression and PI3K-AKT and NF-κB activation. Given that NF-κB directly activates *miR-150* transcription [58], it is conceivable that higher NF-κB activity is responsible for induction of *miR-150* by AS1842856. Moreover, since MYB is a recognized NF-κB target [59], strong NF-κB activation by AS1842856 may also block the observed negative effect of FOXO1 depletion on MYB transcription. 

Importantly, up-regulation of the *miR-150* expression in hematological malignancies is considered a promising therapeutic approach [60,61], warranting further investigations of antitumor effects of FOXO1 inhibitors. 

Although FOXO1 inhibitors did not reach clinical trials yet, numerous preclinical *in vitro* and *in vivo* studies demonstrated their potential efficacy and safety for the treatment of type 2 diabetes. AS1842856, which was discovered to repress FOXO1 dependent transcription of the gluconeogenic enzymes glucose-6 phosphatase and phosphoenolpyruvate carboxykinase, normalizes blood glucose levels in diabetic but not in healthy mice even at concentrations much higher than therapeutic ones [42]. Similarly, AS1842856 potentiated the regeneration of pancreatic β-cells and restored insulin secretion in diabetic mice but did neither increase the number of β-cells nor the insulin concentration in normal mice [62]. Importantly, new selective FOXO1 inhibitors for the treatment of type 2 diabetes have recently been developed by AstraZeneca [63]. 

The participation of FOXOs in the regulation of different biochemical processes in virtually all systems and organs predetermines high repurposing potential of the FOXO1 inhibitors. With respect to cancer therapy, it has been shown that expression of FOXO transcription factors is essential for maintaining a differential blockade in 40% of acute myeloid leukemia (AML) cases [64]. Later, FOXO1 was identified as a critical survival factor in AML1-ETO leukemia, which appeared to be highly sensitive to AS1842856. Importantly, CD34^+^ hematopoietic stem and progenitor cells were more than 10-fold less sensitive to AS1842856 than AML pre-leukemic cells and AML cell lines [65]. Finally, in our recent publication we have shown a high sensitivity of BCP-ALL to the genetic and pharmacological depletion of FOXO1. Most important, by using a *in vivo* BCP-ALL patient-derived xenograft model we demonstrated that AS1842856 induces a significant decrease of tumor load in all critical organ compartments and increases the life span of the animals when administrated at anti-diabetic concentrations [21]. The development of new potent FOXO1 inhibitors could help to increase the efficacy and decrease the toxicity of treatment of FOXO1-dependent tumors including BL. 

Paradoxically, FOXO1 overexpression was also inappropriate for BL maintenance. The antitumor effect of FOXO1 activation was also shown in different B cell lymphomas [13,16,17,18], moreover, for some of them we have provided evidence for a Goldilocks-like (too little is bad as well as too much) behavior of FOXOs [21,23]. Of note, the Goldilocks effect of FOXO1 on cell survival is not restricted to the B cell lineage and has in the meantime been described in other tissues and tumors [66,67]. 

Overall, we have shown that tight regulation of FOXO1 is critical for BL. In particular, our study highlights a role of FOXO1 as an essential regulator of the DZ survival and proliferation program in BL.

## 4. Materials and Methods

### 4.1. Cell Lines 

The BL cell lines (Ramos, BL-41, Namalwa, Daudi, Jiyoye, Raji) and the classical Hodgkin lymphoma (cHL) cell line L-428 were purchased from DSMZ, Braunschweig, Germany. The U-HO1 cell line was generated by one of us (PM) [68]. The culture conditions, analysis of cell line identity, and mycoplasma status were analyzed as described in Appendix A. 

### 4.2. Vectors and Lentiviral Transduction

The cell lines were transduced as described in Appendix A. For expression of the shRNA constructs we used the pRSI12-U6-sh-UbiC-TagRFP-2A-Puro lentiviral vector (BioCat, Heidelberg, Germany). For gene expression we used the SF-LV-cDNA-eGFP vector [69]. The *FOXO1* knockout with help of the CRISPR/Cas9 vector is described in Appendix A. 

### 4.3. Immunoblot and qRT-PCR

Immunoblot and qRT-PCR were done as described in Appendix A. Primer sequences and antibodies are listed in Appendix A. 

### 4.4. Flow Cytometry, Cell Sorting, and Cell Viability Analysis

Growth dynamics of cell lines transduced with lentiviral vectors expressing RFP or GFP were monitored by flow cytometry (FACSCanto II, BD Biosciences, San Jose, CA, USA). For biochemical analysis, RFP^+^ or GFP^+^ cells were sorted using a FACSAria (BD Biosciences) by the Core Facility “Fluorescent Activated Cell Sorting,” Medical Faculty of Ulm, Germany or using the S3e Cell Sorter (Bio-Rad, Hercules, CA, USA). Cell cycle distribution was also measured by flow cytometry using propidium iodide (PI) and cell death was measured by Annexin-V-FITC/PI or Annexin-V-APC/PI staining, as we described previously [20]. CXCR4 surface and intracellular FOXO1 staining was measured using the flow cytometer FACSCanto (BD Biosciences). The sensitivity of the cell lines to the FOXO1 inhibitor was assessed by MTT test. IC50 was calculated by fitting the data points to a nonlinear regression curve using GraphPad Prism (GraphPad Software, San Diego, CA, USA). 

### 4.5. Gene Expression Profiling (GEP) and Gene Set Enrichment Analysis (GSEA)

Total RNA was extracted from sorted cells 4 days post transduction with F1sh or scrambled control and gene expression profiles were analyzed using Human Gene 1.0 ST Affymetrix GeneChip arrays by the Core Facility Genomics (University of Ulm) as we described previously [18]. Probe-level data were obtained using the robust multichip average (RMA) normalization algorithm. The analysis of differentially expressed genes was achieved with help of GeneSifter microarray data analysis software (www.genesifter.net, PerkinElmer; Waltham, MA, USA). For all cell lines and conditions, two biological replicates were analyzed. To identify pathways significantly modulated by FOXO1 depletion we used GSEA (http://software.broadinstitute.org/gsea/index.jsp, 04.07.2016). Microarray data have been deposited in the NCBI GEO (http://www.ncbi.nlm.nih.gov/gds) under the accession number GSE109911. 

### 4.6. Statistical Analysis

The data were analyzed by two-tailed Student’s *t*-test analysis (Microsoft Excel).

## 5. Conclusions

We found that tight regulation of FOXO1 is essential for the survival of BL cell lines and identified maintenance of the DZ B cell survival and proliferation program as the main cause of BL dependency on FOXO1 expression.

## Figures and Tables

**Figure 1 cancers-11-01427-f001:**
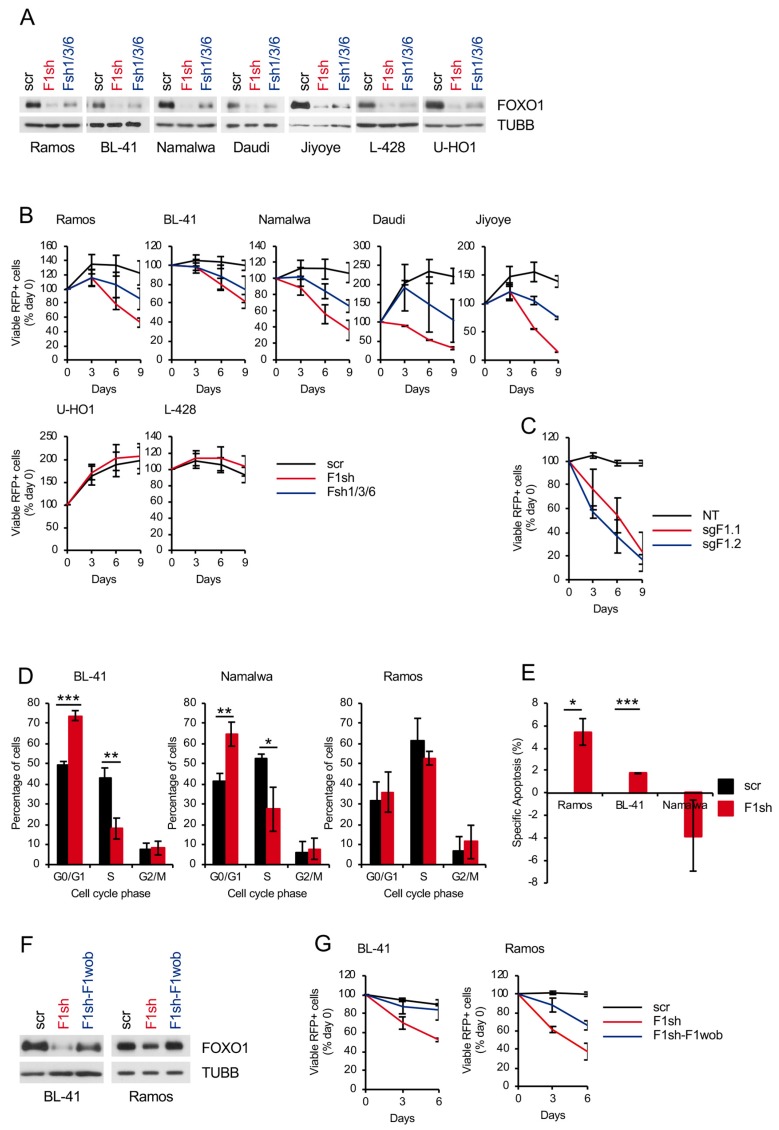
*FOXO1* knockdown negatively regulates proliferation of BL cell lines. (**A**,**B**) BL and cHL cell lines were transduced with lentiviral vectors expressing *FOXO1* shRNA (F1sh) or *FOXO* shRNA1/3/6 (Fsh1/3/6) vs. scrambled (scr) control. (**A**) Knockdown efficiencies of F1sh and Fsh1/3/6 vs. scr control. Transduced cells were selected for 2 days using 4 µg/mL puromycin and FOXO1 expression was analyzed 5–6 days post transduction. Expression of TUBB served as loading control. A representative of 2–3 independent experiments is shown. (**B**) Growth dynamics of transduced BL and cHL cell lines. The percentage of RFP^+^ cells was measured every 3 days using flow cytometry starting from day 4 post transduction. The percentage of RFP^+^ cells at begin of measurements (day 0) was set as 100. Data are shown as mean ± SD (*n* ≥ 3). (**C**) BL cell line Namalwa was transduced with lentiviral vectors co-expressing Cas9 with sgRNAs targeting *FOXO1* (sgF1.1, sgF1.2) or non-targeting (NT) control. For constructs expressing sgRNAs, RFP^+^/FOXO1^−^ cell population was tracked (Appendix A). For NT control, RFP^+^/FOXO1^+^ cell population was tracked, due to the apparent absence of RFP^+^/FOXO1^−^ population in this group. The samples were measured every 3 days by flow cytometry. First measurement was performed 4 days post transduction and the percentage of the indicated cell population was set as 100. Data are shown as mean ± SD (*n* = 2). (**D**) *FOXO1* knockdown inhibits cell cycle progression. BL cell lines expressing F1sh or scr vectors were sorted 4 days post transduction, followed by cell cycle analysis by PI staining. Data are shown as mean percentage of cells in a cell cycle phase ± SD (*n* = 3). The data were analyzed by two-sided T-test. *, *p* < 0.05, **, *p* < 0.01, ***, *p* < 0.001. (**E**) Cell death analysis of BL cell lines expressing F1sh or scr. Transduced cells were sorted 4 days post transduction and incubated in complete medium for 48 h followed by Annexin V-FITC/PI staining. Specific Apoptosis (SA) was calculated as SA (%) = 100 × (A_E_−A_C_)/(100−A_C_), where A_E_ equals the percentage of apoptotic cells in the experimental group and A_C_ equals the percentage of apoptotic cells in the control group. Data are shown as mean ± SD (*n* = 2). The data were analyzed by two-sided T-test. *, *p* < 0.05, ***, *p* < 0.001. (**F**,**G**) BL cell lines were transduced with F1sh, with a vector expressing F1wob and F1sh (F1sh-F1wob), or with scr control. (**F**) Cells were sorted on day 4 and the expression of FOXO1 was analyzed by immunoblot. TUBB served as loading control. A representative of 2 independent experiments is shown. (**G**) The percentage of RFP^+^ cells was measured every 3 days using flow cytometry. First measurement was performed 4–5 days post transduction and the percentage of RFP^+^ cells was set as 100. Data are shown as mean ± SD (*n* = 3).

**Figure 2 cancers-11-01427-f002:**
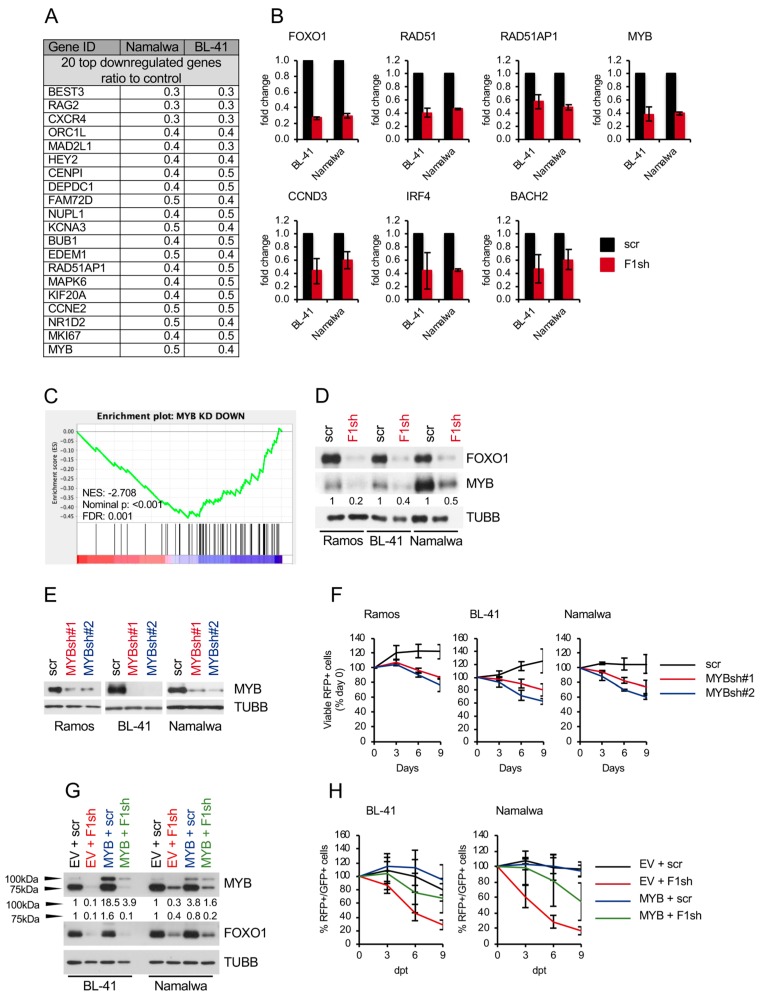
Mechanism of the anti-tumor effect of *FOXO1* knockdown. (**A**) Top 20 genes downregulated after *FOXO1* knockdown > 1.5-fold in both cell lines. BL cell lines expressing F1sh or scr control were harvested on day 4 after transduction for gene expression profiling using Human Gene 1.0 ST Affymetrix GeneChip array. Analysis of raw data was done using Genesifter software (RMA normalization, centered Pearson correlation, coefficient 0.98) (*n* = 2). (**B**) Validation of the GEP. The effect of *FOXO1* knockdown on the expression of indicated genes was validated by qRT-PCR. qRT-PCR data were quantified by the 2^−ΔΔCT^ method. Data are shown as mean ± SD (*n* = 2). The data were analyzed by two-sided T-test. For all genes and cell lines *p* < 0.05. (**C**) GSEA. MYB depletion signature is repressed by *FOXO1* knockdown (Appendix A). Direction of phenotype comparison: “F1sh vs. scr”. Gene signature “MYB KD DOWN” [27] comprises genes downregulated by *MYB* knockdown with *p* < 0.05 (Appendix A) and was applied on set of genes modulated more than 1.5-fold by *FOXO1* knockdown (threshold of 1.5, ANOVA, Benjamini and Hochberg correction, adjusted *p* < 0.05). NES: Normalized enrichment score. FDR: False discovery rate. (**D**) BL cells transduced with F1sh and scr vectors were selected for 2 days using 4 µg/mL puromycin, and MYB expression was analyzed by immunoblot. TUBB served as loading control. A representative of 2 independent experiments is shown. Densitometric quantification of MYB/TUBB protein levels was done with ImageJ software (https://imagej.nih.gov/ij/, RRID: SCR_003070). (**E**,**F**) BL cell lines were transduced with vectors expressing *MYB* shRNAs (MYBsh#1, MYBsh#2) or scr control. (**E**) Cells were selected for 2 days using 4 µg/mL puromycin and MYB expression was analyzed by immunoblot. Expression of TUBB serves as loading control. A representative of 2 or 3 independent experiments is shown. (**F**) Growth dynamics of transduced BL cell lines. The percentage of RFP^+^ cells was measured every 3 days using flow cytometry. First measurement was performed 4 days post transduction and the percentage of RFP^+^ cells was set as 100. Data are shown as mean ± SD (*n* ≥ 3). (**G**,**H**) Expression of MYB rescues BL from inhibition of proliferation induced by *FOXO1* knockdown. BL cell lines expressing MYB or empty vector (EV) were transduced with F1sh vs. scr control. (**G**) GFP^+^/RFP^+^ cells were sorted on day 5 and the expression of FOXO1 and MYB was analyzed by immunoblot. TUBB served as loading control. A representative of 2 independent experiments is shown. Please note that the transduced full length MYB ORF (100 kDa) [35] is also present in not transduced cells. Densitometric quantification of 75 kDa MYB/TUBB and 100 kDa MYB/TUBB protein levels was done with ImageJ software (https://imagej.nih.gov/ij/, RRID: SCR_003070). (**H**) The percentage of GFP^+^/RFP^+^ cells was measured every 3 days using flow cytometry. First measurement was performed 4–5 days post transduction and the percentage of RFP^+^ cells was set as 100. Data are shown as mean ± SD (*n* = 3).

**Figure 3 cancers-11-01427-f003:**
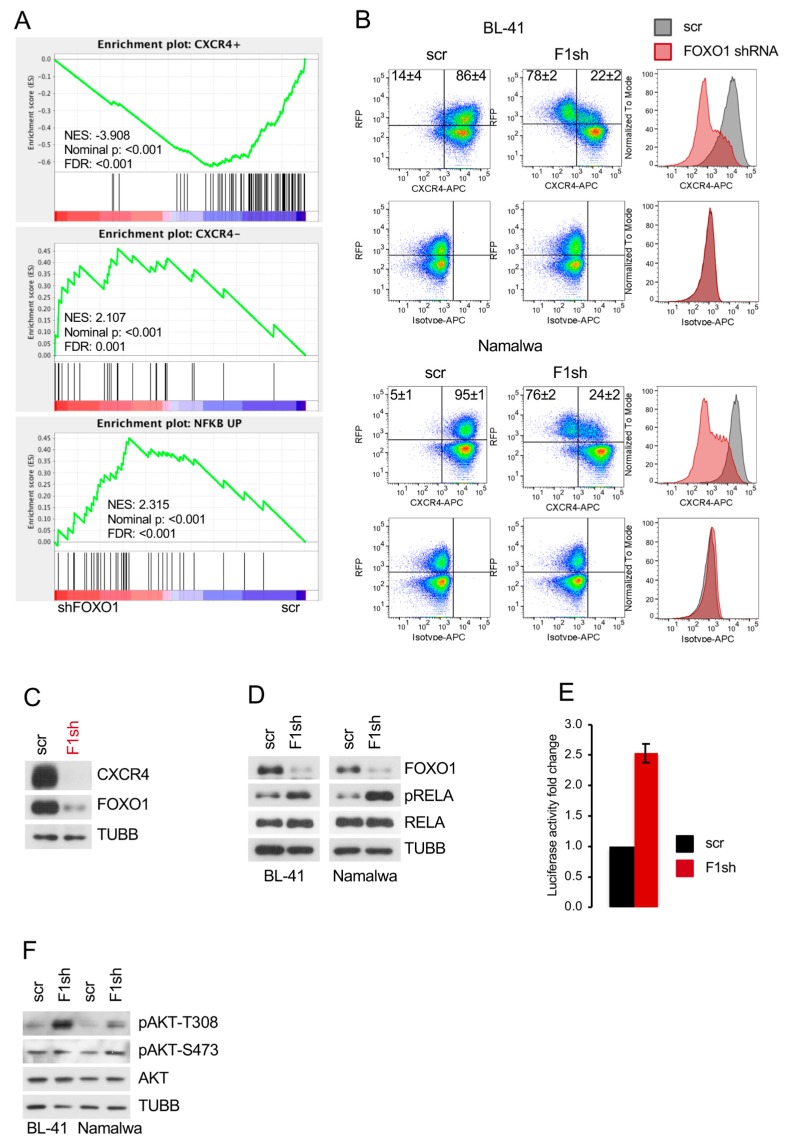
*FOXO1* knockdown induces loss of the DZ program and PI3K-AKT and IKK-NF-κB activation. (**A**) GSEA. Gene signatures comprising genes differentially expressed genes in CXCR4^+^ centroblasts vs. CXCR4^−^ centrocytes [36] and gene signature comprising genes differentially expressed in Ramos cell line transfected with constitutively active CA-IKK2 vs. EV control [40] were applied on set of genes modulated more than 1.5 fold by *FOXO1* knockdown (threshold of 1.5, ANOVA, Benjamini and Hochberg correction, adjusted *p* < 0.05). Direction of phenotype comparison: “F1sh vs. scr” (Appendix A). (**B**) F1sh downregulates CXCR4. BL cell lines expressing F1sh or scr control were stained with CXCR4-APC or isotype control 7 days post transduction. Dot-plots show percentages of CXCR4^+^/RFP^+^ and CXCR4^−^/RFP^+^ cells. For histograms, only transduced RFP^+^ cells were included. Data are shown as mean ± SD (*n* = 3). (**C**) BL-41 cell line expressing F1sh or scr control was FACS sorted 4 days post transduction and CXCR4 levels were analyzed by immunoblot. TUBB served as loading control. A representative of 2 independent experiments is shown. (**D**) *FOXO1* knockdown activates IKK. BL cell lines expressing F1sh or scr control were sorted 4 days post transduction and phosphorylation status of RELA^S536^ (IKKs target), and total RELA were analyzed by immunoblot. TUBB served as loading control. A representative of 2 independent experiments is shown. (**E**) Luciferase reporter assay using the Namalwa cell line stably expressing a NFκB-dependent luciferase reporter (3× κB.luc) [41]. Cells were transduced with F1sh followed by FACS sorting for RPF 4 days post transduction and cell lysis. Luminescence was measured as described in Appendix A. Data are shown as mean ± SD (*n* = 3). (**F**) BL cell lines expressing F1sh or scr control were FACS sorted 7 days post transduction and pAKT^T308^ and pAKT^S473^ levels were analyzed by immunoblot. TUBB served as loading control. A representative of 2 independent experiments is shown.

**Figure 4 cancers-11-01427-f004:**
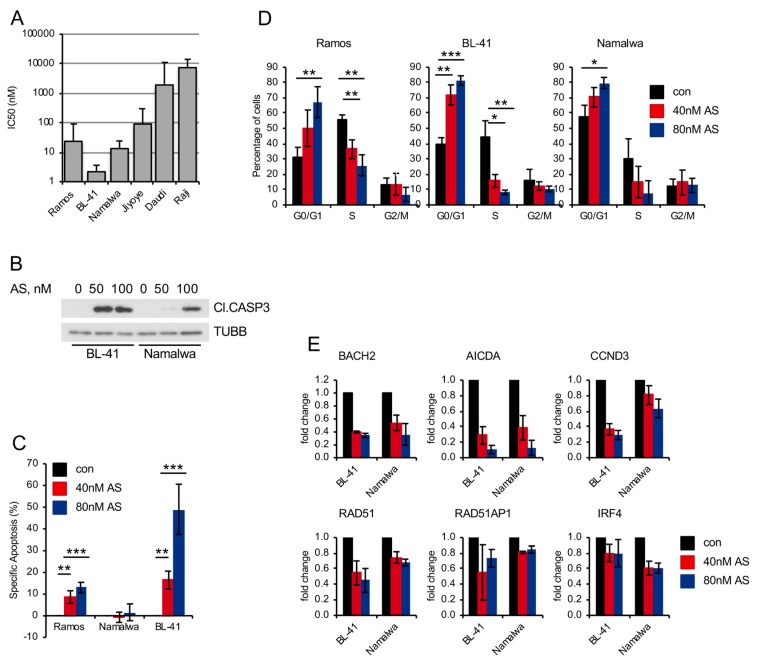
AS1842856 treatment is toxic for BL cell lines. (**A**) Sensitivities of BL cell lines to FOXO1 inhibitor AS1842856 (AS). Cells were exposed to increasing concentrations of AS for 5 days. Cell viability was assessed using MTT assay and IC50 values were calculated using GraphPad Prism software. Data are shown as mean ± SD (*n* = 3). (**B**) Cleaved CASP3 was detected in BL cell lines treated with AS or DMSO for 4 days using immunoblot. TUBB served as loading control. A representative of 3 independent experiments is shown. (**C**) AS induces apoptosis in BL cell lines. Cells were treated for 4 days with AS or DMSO, followed AnnexinV-FITC/PI staining. Data are shown as mean of specific apoptosis ± SD (*n* = 2). The data were analyzed by two-sided T-test. **, *p* < 0.01, ***, *p* < 0.001. (**D**) AS inhibits cell cycle progression. Cells were incubated in the presence of AS or DMSO for 4 days, followed by fixation and PI staining. After FACS measurement percentage of cells in different cell cycle phases was assessed and is shown as mean ± SD (*n* = 3). The data were analyzed by two-sided T-test. *, *p* < 0.05, **, *p* < 0.01, ***, *p* < 0.001. (**E**) Downregulation of FOXO1 target genes after treatment with AS compared to DMSO-treated cells. Treated cells were harvested after 24 h and RNA expression levels were analyzed by qRT-PCR. qRT-PCR data were quantified by the 2^−ΔΔCT^ method. Data are shown as mean ± SD (*n* = 3). The data were analyzed by two-sided T-test. For all genes and cell lines *p* < 0.05 with exception of Namalwa + 40 nM AS CCND3 *p* = 0.051; BL-41 + 40 nM AS RAD51AP1 *p* = 0.136.

**Figure 5 cancers-11-01427-f005:**
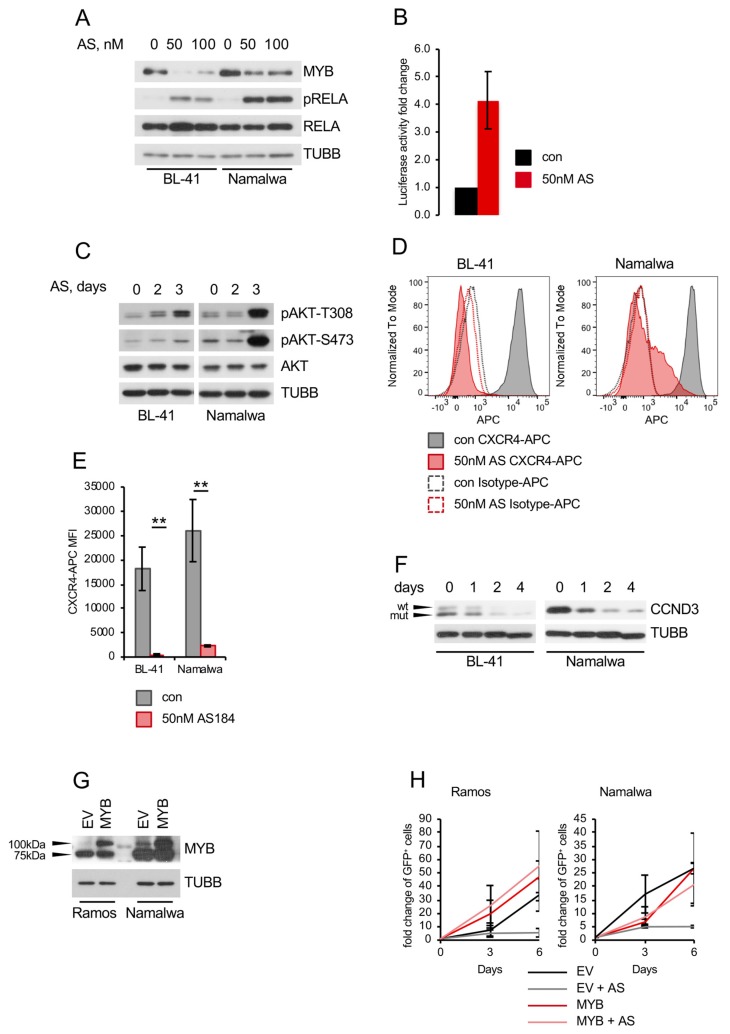
AS1842856 treatment mimics genetic FOXO1 inhibition. (**A**) Immunoblot showing reduced MYB and increased pRELA^S536^ levels after AS treatment for 4 days compared to DMSO-treated cells. TUBB served as loading control. Same loading control as in Figure 4B was used, since the same membrane was probed. A representative of 3 independent experiments is shown. (**B**) Luciferase reporter assay using the Namalwa cell line stably expressing a NF-κB-dependent luciferase reporter (3× κB.luc) [41]. Cells were treated with 50 nM AS for 3 days followed by cell lysis. Luminescence was measured as described in Appendix A. Data are shown as mean ± SD (*n* = 3). (**C**) Immunoblot showing pAKT^T308^ and pAKT^S473^ levels after 2 and 3 days of AS treatment compared to DMSO-treated cells. TUBB served as loading control. A representative of 2 independent experiments is shown. (**D**,**E**) AS downregulates CXCR4. BL cell lines were treated for 4 days with 50 nM AS or DMSO. Treated cells were stained with CXCR4-APC or isotype control. (**D**) FACS histogram plots of stained cells. Data are shown as mean ± SD (*n* = 3). (**E**) Mean fluorescence intensity (MFI) of CXCR4-APC signal of AS-treated DMSO-treated cells. Data are shown as mean ± SD (*n* = 3). The data were analyzed by two-sided T-test. **, *p* < 0.01. (**F**) AS decreases CCND3 protein level. BL cell lines were treated for up to 4 days with AS or DMSO and CCND3 levels were analyzed by immunoblot. Arrows indicate wild type and mutated version of CCND3 in BL-41. TUBB served as loading control. A representative of 2 independent experiments is shown. (**G**,**H**) MYB expression rescues BL cell lines from the toxic effect of AS. BL cell lines were transduced with lentiviral vectors expressing MYB vs. empty vector (EV) control. (**G**) Transduced cells were FACS sorted 4–5 days post transduction and MYB expression was analyzed. TUBB served as loading control. A representative of 2 independent experiments is shown. (**H**) Cells were treated with 100 nM AS vs. DMSO control. Percentages of GFP^+^ cells (by flow cytometry) and the total number of live cells (by cell counting) were measured at indicated time points. Data are shown as fold changes normalized to the initial number of live GFP^+^ cells. The number of live GFP^+^ cells was calculated as *n* X % GFP^+^ cells/100, where *n* is the number of live cells per well and % GFP^+^ is the percentage of GFP^+^ cells. Data are shown as mean ± SD (*n* ≥ 3).

**Figure 6 cancers-11-01427-f006:**
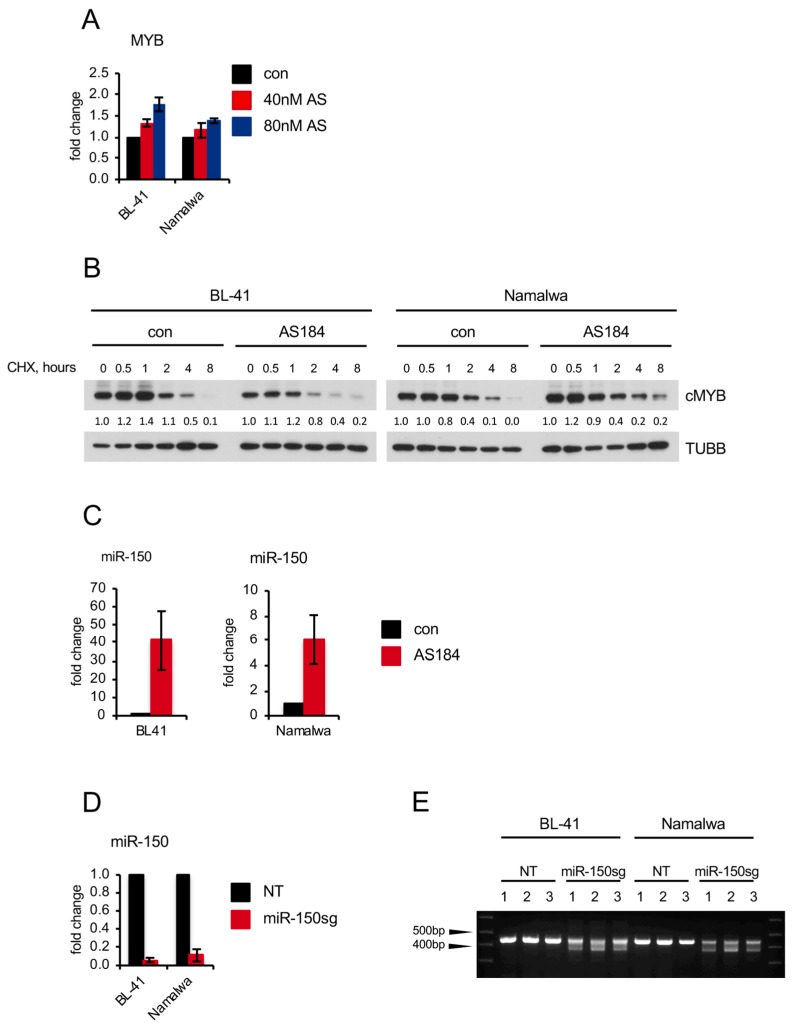
AS1842856 regulates MYB through upregulation of miR-150 (**A**) AS does not reduce MYB RNA levels. BL cells were treated with AS or DMSO. Treated cells were harvested after 24 h and RNA expression levels were analyzed by qRT-PCR. qRT-PCR data were quantified by the 2^−ΔΔCT^ method. Data are shown as mean ± SD (*n* = 3). (**B**) AS does not induce MYB protein degradation. BL cells were treated with 10 µg/mL CHX and harvested 0–8 h later. Expression of MYB was analyzed by immunoblot. Expression levels of TUBB served as loading control. A representative of 2 independent experiments is shown. Densitometric quantification of MYB/TUBB protein levels was done with ImageJ software (https://imagej.nih.gov/ij/, RRID: SCR_003070). (**C**) AS induces miR-150 transcription. BL cells were treated with AS or DMSO. Treated cells were harvested after 3 days, followed by miRNA isolation and miRNA cDNA generation as described in Appendix A. miRNA expression levels were analyzed by qRT-PCR. qRT-PCR data were quantified by the 2^−ΔΔCT^ method. Data are shown as mean ± SD (*n* = 3). (**D**–**F**) BL cell lines were transduced with lentiviral vectors expressing a miR-150 CRISPR/Cas9 knockout construct or a non-targeting control (NT) and FACS sorted. (**D**) miRNA was isolated and miRNA cDNA was generated 4 days post transduction as described in Appendix A. miRNA expression levels were analyzed by qRT-PCR. qRT-PCR data were quantified by the 2^−ΔΔCT^ method. Data are shown as mean ± SD (*n* = 3). (**E**) miR-150 knockout construct is functional. Genomic DNA was isolated 4 days post transduction and the miR-150 locus was amplified as described in Appendix A (*n* = 3). (**F**) miR-150 knockout rescues MYB levels in BL cell lines treated with AS. Cells expressing miR-150sg or NT con were treated with AS for 7 days. Expression of MYB was analyzed by immunoblot. Expression levels of TUBB served as loading control. A representative of 2 independent experiments is shown. Densitometric quantification of MYB/TUBB protein levels was done with ImageJ software (https://imagej.nih.gov/ij/, RRID: SCR_003070).

**Figure 7 cancers-11-01427-f007:**
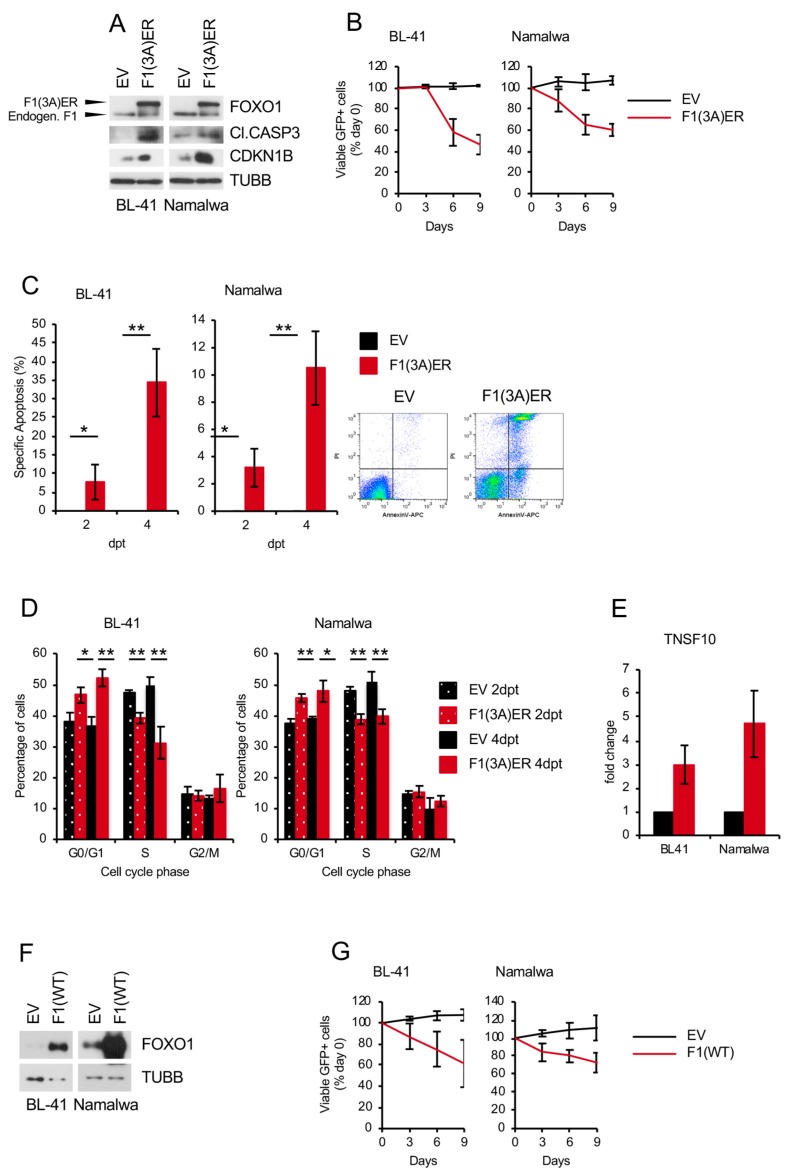
FOXO1 induces growth inhibition in BL cell lines (**A–E**) BL cell lines were transduced with lentiviral vectors expressing constitutively active FOXO1(3A) ER (F1ER) vs. empty vector (EV) control. (**A**) Transduced cells were FACS sorted and lysed 2 days post induction with 100 nM 4-OHT. Expression levels of FOXO1, cleaved CASP3 and CDKN1B were analyzed by immunoblot. TUBB served as loading control. A representative of 3 independent experiments is shown. (**B**) Growth dynamics of transduced BL cell lines. The percentage of GFP^+^ cells was measured every 3 days using flow cytometry. First measurement was performed 4 days post transduction and the percentage of GFP^+^ cells was set as 100 %. Data are shown as mean ± SD (*n* ≥ 3). (**C**) Cell death analysis of BL cell lines. Transduced cells were FACS sorted, followed by AnnexinV-APC/PI staining 2- and 4-days post induction with 100 nM 4-OHT. After FACS measurement, Specific Apoptosis (SA) was calculated as SA (%) = 100 × (AE−AC)/(100−AC), where AE equals the percentage of apoptotic cells in the experimental group and AC equals the percentage of apoptotic cells in the control group. Data are shown as mean ± SD (*n* = 3). The data were analyzed by two-sided T-test. *, *p* < 0.05, **, *p* < 0.01. Representative dot-plot images are shown. (**D**) FOXO1 overexpression inhibits cell cycle progression. Transduced cells were FACS sorted, followed cell cycle analysis by PI staining 2- and 4-days post induction with 100 nM 4-OHT. Data are shown as mean percentage of cells in a cell cycle phase ± SD (*n* = 3). The data were analyzed by two-sided T-test. *, *p* < 0.05, **, *p* < 0.01. (**E**) qRT-PCR analysis of FOXO1 target gene TNSF10. Transduced cells were FACS sorted and RNA was isolated 2 days post induction with 100 nM 4-OHT. qRT-PCR data were quantified by the 2^−ΔΔCT^ method. Data are shown as mean ± SD (*n* = 3). (**F**,**G**) BL cell lines were transduced with lentiviral vectors expressing wildtype FOXO1 (F1(WT)) vs. EV control. (**F**) Transduced cells were FACS sorted 4–5 days post transduction. Expression levels of FOXO1 were analyzed by immunoblot. TUBB served as loading control. A representative of 2 independent experiments is shown. (**G**) Growth dynamics of transduced BL cell lines. The percentage of GFP^+^ cells was measured every 3 days using flow cytometry. First measurement was performed 4–5 days post transduction and the percentage of GFP^+^ cells was set as 100 %. Data are shown as mean ± SD (*n* = 3).

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
