# Peer review of "FOXO1 Confers Maintenance of the Dark Zone Proliferation and Survival Program and Can Be Pharmacologically Targeted in Burkitt Lymphoma"

_cancers, 2019, doi:10.3390/cancers11101427_

Round 1

Reviewer 1 Report

This is a very interesting manuscript demonstrating that FOXO1 expression levels are critical to BL cell proliferation and that both upregulation and downregulation negatively affect cancer cell survival.

l suggest to the authors to better address the pharmacological part of the study, and particularly the mechanisms leading to miRNA upregulation and the miRNA mediated regulation of the gene. Please also discuss on the safety and tolerability of the drug, and the potential manipulation of miRNAs as a targeting approach.

Author Response

We are thankful to the reviewer for the valuable comments. We addressed all questions asked by the reviewer.

This is a very interesting manuscript demonstrating that FOXO1 expression levels are critical to BL cell proliferation and that both upregulation and downregulation negatively affect cancer cell survival.

Q1. l suggest to the authors to better address the pharmacological part of the study, and particularly the mechanisms leading to miRNA upregulation and the miRNA mediated regulation of the gene. Please also discuss on the safety and tolerability of the drug, and the potential manipulation of miRNAs as a targeting approach.

A1. Thank you for your suggestion. We included the following text in the discussion of the manuscript:

Pages 19-20.

The most obvious differences between shRNA-mediated and pharmacological FOXO1 inactivation were the inability of AS1842856 to repress MYB at mRNA level. Interestingly, repression of MYB protein expression was due to induction of miR-150 transcription. These discrepancies might be explained all by differences in the mechanisms of action. RNA interference decreases the FOXO1 levels, whereas AS1842856 does not modulate FOXO1 expression or localization, instead, AS1842856 binds to the transactivation domain and thereby bona fide interferes with the FOXO1 transactivation activity [42]. It is conceivable that not all interactions of FOXO1 with other proteins might be blocked by AS1842856, moreover, the binding specificity of the new structures of AS1842856 might be different. Importantly, AS1842856 binds, although to a lesser extent, to other members of the FOXO family, the effects of whose are only partially redundant [42]. Inhibition of other FOXOs might explain the generally stronger effects of pharmacological in comparison to genetic FOXO1 inhibition in regard to gene expression and PI3K-AKT and NF-κB activation. Given that NF-κB directly activates miR-150 transcription [58], it is conceivable that higher NF-κB activity is responsible for the induction of miR-150 by AS1842856. Moreover, since MYB is a recognized NF-κB target [59], strong NF-κB activation by AS1842856 may also block the observed negative effect of FOXO1 depletion on MYB transcription.

Importantly, up-regulation of the miR-150 expression in hematological malignancies is considered a promising therapeutic approach [60,61], warranting further investigations of antitumor effects of FOXO1 inhibitors.

Although FOXO1 inhibitors did not reach clinical trials yet, numerous preclinical in vitro and in vivo studies demonstrated their potential efficacy and safety for the treatment of type 2 diabetes. AS1842856, which was discovered to repress FOXO1 dependent transcription of the gluconeogenic enzymes glucose-6 phosphatase and phosphoenolpyruvate carboxykinase, normalizes blood glucose levels in diabetic but not in healthy mice even at concentrations much higher than therapeutic ones [42]. Similarly, AS1842856 potentiated the regeneration of pancreatic β-cells and restored insulin secretion in diabetic mice, but did neither increase the number of β-cells nor the insulin concentration in normal mice [62]. Importantly, new selective FOXO1 inhibitors for the treatment of type 2 diabetes have recently been developed by AstraZeneca [63].

The participation of FOXOs in the regulation of different biochemical processes in virtually all systems and organs predetermines high repurposing potential of the FOXO1 inhibitors. With respect to cancer therapy, it has been shown that expression of FOXO transcription factors is essential for maintaining a differential blockade in 40 % of acute myeloid leukemia (AML) cases [64]. Later, FOXO1 was identified as a critical survival factor in AML1-ETO leukemia, which appeared to be highly sensitive to AS1842856. Importantly, CD34+ hematopoietic stem and progenitor cells were more than 10-fold less sensitive to AS1842856 than AML pre-leukemic cells and AML cell lines [65]. Finally, in our recent publication we have shown a high sensitivity of BCP-ALL to the genetic and pharmacological depletion of FOXO1. Most important, by using a in vivo BCP-ALL patient-derived xenograft model we demonstrated that AS1842856 induces a significant decrease of tumor load in all critical organ compartments and increases the life span of the animals when administrated at anti-diabetic concentrations [21]. The development of new potent FOXO1 inhibitors could help to increase the efficacy and decrease the toxicity of treatment of FOXO1-dependent tumors including BL.

Reviewer 2 Report

Gehringer et al., presented the functional role of nuclear localized FOXO1 in Burkitt lymphoma at germinal center B cells using RNAi and CRISPR against FOXO1 along with targeting FOXO1 target genes, such as MYB. They found that the KD of FOXO1 repressed the proliferation of cancer cell lines that they used (not all). In addition, they also showed that FOXO1 inhibitor,  AS1842856 can induce the cell death and growth arrest in Bcell lymphoma cell lines. Although the results were solid and well demonstrated, some results (the role of FOXO1 on B cell lymphoma of germinal center B cells were previously presented in the other group (Kabrani et al., 2018, Blood). There are some analyses that they could improve their study as below. 

[Comments]

In Fig1 C, they also showed the greater decrease of RFP+ cells in response to sgRNAs that target FOXO1 than in response to shRNAs. They need to show how the level of FOXO1 was changed by CRISPR/Cas9. In Fig1 D and E, they showed the G1 arrest by FOXO1 KD in BL-41 and Namalwa cell lines but not in Ramos. However, the Apoptotic events were observed in Ramos and BL-41 but not in  Namalwa. Is there any explanation for the inconsistency on the same cell lines? In Fig2, they  could also compare the level of targets in the cells transfecting CRISPR/Cas9 and sgRNAs to remove any siRNA off-targets.

Author Response

We are thankful to the reviewer for the valuable comments. We addressed all questions asked by the reviewer and included the requested experiment.

Gehringer et al., presented the functional role of nuclear localized FOXO1 in Burkitt lymphoma at germinal center B cells using RNAi and CRISPR against FOXO1 along with targeting FOXO1 target genes, such as MYB. They found that the KD of FOXO1 repressed the proliferation of cancer cell lines that they used (not all). In addition, they also showed that FOXO1 inhibitor, AS1842856 can induce the cell death and growth arrest in B cell lymphoma cell lines. Although the results were solid and well demonstrated, some results (the role of FOXO1 on B cell lymphoma of germinal center B cells were previously presented in the other group (Kabrani et al., 2018, Blood). There are some analyses that they could improve their study as below. 

Q2. In Fig1 C, they also showed the greater decrease of RFP+ cells in response to sgRNAs that target FOXO1 than in response to shRNAs. They need to show how the level of FOXO1 was changed by CRISPR/Cas9.

A2. The downregulation of FOXO1 by CRISPR/Cas9 knockout constructs is shown in Supplementary Figure 2B and in new Supplementary Figure 2D.

Q2. In Fig1 D and E, they showed the G1 arrest by FOXO1 KD in BL-41 and Namalwa cell lines but not in Ramos. However, the Apoptotic events were observed in Ramos and BL-41 but not in Namalwa. Is there any explanation for the inconsistency on the same cell lines?

A2. In BL-41 we observed a Specific Apoptosis rate of 1.8%. Although significant, this is not a relevant increase of apoptosis. In Ramos, we detected a slightly higher Specific Apoptosis of 5.4%. In addition, it is conceivable that FOXO1 knockdown induces a cell cycle unspecific arrest.

Q3. In Fig2, they could also compare the level of targets in the cells transfecting CRISPR/Cas9 and sgRNAs to remove any siRNA off-targets.

A3. In this manuscript, we demonstrated that FOXO1 expression in BL is critical for maintenance of the DZ program and that FOXO1 knockdown results in downregulation of critical DZ factors CXCR4 and MYB. Therefore, we analysed the effect of the CRISPR/Cas9 FOXO1 knockout constructs on expression of CXCR4 and MYB proteins. Genomic editing of FOXO1 decreased CXCR4 and MYB expression (new Supplemental Figure 2D).

The following text is added:

Page 8:

Finally, to exclude the possibility that the observed downregulation of GC DZ genes represents an shRNA off-target effect we corroborated the results of FOXO1 knockdown by FOXO1 genome editing. Using immunoblot we found that both sgRNAs downregulated the expression of CXCR4 and MYB in the BL cell line Namalwa (Figure S2D).

Round 2

Reviewer 2 Report

All initial concerns and suggestions were addressed properly. Now, the revised manuscript was significantly improved.